# PeerJ

# A large-scale field study examining effects of exposure to clothianidin seed-treated canola on honey bee colony health, development, and overwintering success

G. Christopher Cutler[1], Cynthia D. Scott-Dupree[2], Maryam Sultan[2], Andrew D. McFarlane[2] and Larry Brewer[3]

[1] Department of Environmental Sciences, Faculty of Agriculture, Dalhousie University, Truro, NS, Canada
[2] School of Environmental Sciences, University of Guelph, Guelph, ON, Canada
[3] Smithers Viscient, Carolina Research Center, Snow Camp, NC, USA

Corresponding author
G. Christopher Cutler, chris.cutler@dal.ca

## ABSTRACT

In summer 2012, we initiated a large-scale field experiment in southern Ontario, Canada, to determine whether exposure to clothianidin seed-treated canola (oil seed rape) has any adverse impacts on honey bees. Colonies were placed in clothianidin seed-treated or control canola fields during bloom, and thereafter were moved to an apiary with no surrounding crops grown from seeds treated with neonicotinoids. Colony weight gain, honey production, pest incidence, bee mortality, number of adults, and amount of sealed brood were assessed in each colony throughout summer and autumn. Samples of honey, beeswax, pollen, and nectar were regularly collected, and samples were analyzed for clothianidin residues. Several of these endpoints were also measured in spring 2013. Overall, colonies were vigorous during and after the exposure period, and we found no effects of exposure to clothianidin seed-treated canola on any endpoint measures. Bees foraged heavily on the test fields during peak bloom and residue analysis indicated that honey bees were exposed to low levels (0.5–2 ppb) of clothianidin in pollen. Low levels of clothianidin were detected in a few pollen samples collected toward the end of the bloom from control hives, illustrating the difficulty of conducting a perfectly controlled field study with free-ranging honey bees in agricultural landscapes. Overwintering success did not differ significantly between treatment and control hives, and was similar to overwintering colony loss rates reported for the winter of 2012–2013 for beekeepers in Ontario and Canada. Our results suggest that exposure to canola grown from seed treated with clothianidin poses low risk to honey bees.

## INTRODUCTION

The neonicotinoid class of insecticides—which includes imidacloprid, acetamiprid, clothianidin, thiamethoxam, thiacloprid, dinotefuran and nitenpyram—are considered an important tool for pest management in many agricultural systems. As of 2006, this

insecticide class accounted for approximately $US 1.56 billion worldwide, representing nearly 17% of the global insecticide market (*Jeschke & Nauen, 2008*). When first introduced in the mid-1990s, neonicotinoids were well-received by pesticide regulators, ecotoxicologists, and farmers, owing to their novel mode of action, efficacy against multiple pests, and selectivity for insects over vertebrates (*Jeschke & Nauen, 2008*; *Matsuda et al., 2001*). In addition, the systemic activity of neonicotinoids allows them to be applied to soil or seeds at low rates, providing protection to crops in their more vulnerable early stages. This reduces the number of foliar insecticide applications required, which are applied at much greater application rates and generally pose more hazard to non-target organisms.

There is much concern, however, regarding potential risks of neonicotinoids to pollinators, mainly bees (Apoidea). Several neonicotinoids are highly toxic to bees (*Iwasa et al., 2004*; *Scott-Dupree, Conroy & Harris, 2009*) and mitigation measures are needed to minimize pollinator exposure where identified hazards may occur. For example, for foliar applications of compounds in the nitroguanidine class of neonicotinoids (imidacloprid, thiamethoxam, clothianidin, and dinotefuran), there are warnings on product labels in North America not to apply or allow them to drift on to flowering crops or weeds if bees are foraging in the treated area. To minimize exposure to contaminated dust generated during the planting of neonicotinoid treated seeds, which can result in bee-kill incidents (*Cutler, Scott-Dupree & Drexler, 2014*), efforts are being made to improve the seed treatment process, modify planting equipment, and encourage best management practices among growers and beekeepers to reduce pollinator risk from exposure to neonicotinoid contaminated dust from treated seed (*Nuyttens et al., 2013*; *Health Canada Pest Management Regulatory Agency, 2013*).

There is perhaps more debate regarding potential risks to bees through feeding on nectar or pollen from plants grown from seed treated with neonicotinoids. Several studies have found that neonicotinoids can cause various adverse chronic/sublethal effects on honey bees (*Apis* spp.) and bumble bees (*Bombus* sp.). These studies have been important in demonstrating different ways toxicity can occur, and the potential hazards neonicotinoids pose to pollinators. Some have argued, however, that such studies have used unrealistic exposure scenarios (*Campbell, 2013*; *Cresswell, 2013*; *Cresswell & Thompson, 2012*; *EFSA, 2012b*; *Walters, 2013*), either subjecting bees to doses that are higher than those typically experienced in field (*Gill, Ramos-Rodriguez & Raine, 2012*; *Henry et al., 2012*), or subjecting bees in the laboratory exclusively to food spiked with neonicotinoids for prolonged periods (*Whitehorn et al., 2012*). On the other hand, semi-field (field cage) and field studies have found that individual bees and colonies are not adversely impacted when foraging on neonicotinoid seed-treated crops (*Cutler & Scott-Dupree, 2007*; *Cutler & Scott-Dupree, 2014*; *Nguyen et al., 2009*; *Pilling et al., 2013*; *Pohorecka et al., 2012*; *Schmuck & Keppler, 2003*; *Schmuck et al., 2001*; *Schneider et al., 2012*; *Tasei, Ripault & Rivault, 2001*; *Thompson et al., 2013*).

Clothianidin is used on millions of hectares of canola (*Brassica napus* L.) in western Canada and elsewhere, mainly to provide protection against early-season defoliators such as flea beetles (*Phyllotreta* spp.). There is concern by some scientists, beekeepers, legislators,

and members of the general public that bees foraging on clothianidin seed-treated canola will suffer acute or chronic effects that compromise colony health. Here we present results of a large-scale field study done in 2012–2013 in southern Ontario, Canada, undertaken to determine whether or not exposure to clothianidin seed-treated canola has any adverse impacts on honey bees (*Apis mellifera* L). We examined numerous colony endpoints before, during, and after treatment exposure in the field.

## MATERIALS AND METHODS

This research was conducted in accordance with the Organization for Economic Cooperation and Development Principles of Good Laboratory Practice (*EPA, 1989*; *OECD, 1999*). The experimental design was developed by GCC, CDSD, LB, in consultation with personnel from Bayer CropScience, the Health Canada Pest Management Regulatory Agency, and the United States Environmental Protection Agency. No claim of confidentiality is made for any information contained in this study on the basis of its falling within the scope of the Federal Insecticide, Fungicide, and Rodenticide Act, FIFRA Section 10(d)(1)(A), (B), or (C).

### Seed treatment

Clothianidin (CAS No.: 205510-53-8) was applied to canola seed as Prosper FX® formulation (20.4 % clothianidin, 0.5% trifloxystrobin, 3.6% carbathiin and 0.4% metalaxyl) at the Bayer CropScience Seed Technology Center (Research Triangle Park, NC). Seed was treated at the target label rate of 1,400 ml Prosper per 100 kg of seed. An equal amount of seed was treated with a control formulation that contained trifloxystrobin, carbathiin, and metalaxyl at their registered label rates, but did not contain clothianidin. Seed was shipped to the Bayer CropScience Canada Rockwood Research Farm (Rockwood, ON) and stored in plastic bins (separate seed bin for each treatment) at temperatures that ranged from 3.9 to 27.4 °C. Subsequent analysis of treated seed confirmed that the targeted seed treatment rate was met, at 91% of the nominal treatment rate (maximum allowed on the label), which is within the acceptable range of error of the analytical method.

### Field sites and planting

Fields under the ownership of cooperating farmers were used in this experiment and their consent was granted to access study sites, and to apply pesticides and fertilizers. Application of pesticides and fertilizer complied with all government and manufacturer regulations.

Ten fields in southwest Ontario, Canada, suitable for growing spring canola were chosen. Fields were in Brant (1), Oxford (1), Waterloo (5), Wellington (2), and Wentworth (1) counties within an area of approximately 60 × 65 km. To our knowledge, which involved consultation with growers throughout the region and ground-truthing the area around the test sites, no other canola was being grown within foraging distance of our experimental fields. Fields were located a minimum of 10 km (6 mi) apart and in the previous 12 months had not received applications of neonicotinoids such as clothianidin (Poncho, Titan, and Prosper), imidacloprid (Admire, Gaucho, Alias, Grapple, and Stress

Shield), thiacloprid (Calypso), acetamiprid (Assail), or thiamethoxam (Cruiser, Actara, and Helix). Fields that had been planted with wheat the previous year were chosen to minimize the possibility that selected study fields had received neonicotinoid treatments in the previous year. In southern Ontario, only a very small percentage of wheat is grown from seed treated with neonicotinoids and these wheat fields would have been planted the previous fall (2010). Thus, the time since any previous neonicotinoid treatments had been made on test fields to the start of the study was at least 1.5 years.

All sites received 400 kg/ha of fertilizer (28% N, 5.6% P, 7% K, 8.4% S), applied with a broadcast spreader and were prepared for planting with a McFarlane Reel Disk. Five field sites selected at random were planted as "control" fields and the other five sites were planted as "clothianidin" (treatment) fields. All fields were planted over three days on 10–12 May 2012. Each site was planted with approximately 2 ha (5 acres; range = 2.00–2.19 ha) canola according to local agronomic practices. The seeder was calibrated to deliver 5.6 kg canola seed/ha (5 lbs/acre), which ensured a high number of plants and an abundance of blooms on which bees could forage. All sites were treated equally with Liberty® 200SN (glufosinate ammonium) for weed control and Decis® 5EC (deltamethrin) for early season flea beetle control. The interval between the last Decis 5EC application and the time when hives were first placed in the test fields was at least 30 days.

## Colony preparation and management

Prior to placement in canola plots on the study sites, 44 honey bee colonies were maintained at a spring apiary located at the Arkell Agricultural Research Station (N43°31′3.8″; W80°10′4.9″) and under the management of the Honey Bee Research Facility (HBRF), University of Guelph, Guelph, Ontario. Forty of these colonies were used in the study, and four were maintained as spares. Each colony consisted of a single brood chamber measuring 24 cm (9⅝″) deep, containing 9 frames and one follower board to replace the 10th frame. The follower board in the brood chamber maintained the 10-frame spacing typical of commercial colonies, while facilitating the frequent colony assessments conducted during the study by allowing more working space in the brood chamber. A single shallow empty honey super, measuring 16.5 cm (6⅝″) deep and containing 9 frames with plastic foundation, was placed above the brood chamber. Queen bees were provided by HBRF and all were of the same lineage and approximately the same age. A queen excluder was placed between the brood chamber and honey super to confine the queen to the brood chamber. Colonies were adjusted for strength, as necessary, prior to being moved to the canola fields. The strength adjustments established similar quantities of food stores (pollen and nectar), sealed brood, and adults in each colony.

Colonies also were assessed for presence of Varroa mite (*Varroa destructor*), tracheal mite (*Acarapis woodi*), American foulbrood (AFB; *Paenibacillus larvae*), European foulbrood (EFB; *Melissococcus plutonius*), *Nosema* spp. (*N. apis* and/or *N. ceranae*), and chalkbrood before placement in canola fields (*Shimanuki & Knox, 2000*). Hives infected with diseases as determined during the initial hive assessments were not used in the study. Disease and parasite analyses were conducted again after removal from canola, and during

spring 2013 assessments. Colonies were treated with Oxytet-25 (Oxytetracycline HCL; Medivet Pharmaceuticals Ltd., High River, AB) mixed with powdered sugar in early spring of 2012 and in early October 2012 to prevent EFB/AFB. To treat for Varroa mites, hives received Apivar® (Amitraz; Medivet Pharmaceuticals Ltd., High River, AB) strips in early October of 2012. These medications and acaricides were applied before and after the field (canola) phase of the study. No medications were applied to hives while they were in test fields. All study colonies received equal mite/disease treatment, even when threshold levels of these pests were not present in some colonies. Pest and disease status of colonies were assessed and recorded during hive assessments, which were conducted during 2012 on day-4, 7, 14, 21, 42, 63, and 84, and during spring 2013 assessment.

Honey supers were added to or removed from colonies as needed (i.e., removed when they were full of honey). Brood boxes and honey supers were weighed and labeled to facilitate accurate colony-component cross-referencing and accurate assessment of productivity by weight.

## Colony transport and placement

A 7 × 7 m clearing was mowed in the middle of each canola field to accommodate four colonies. The central clearing was vehicle accessible via a laneway running from the edge of the field on one side to the clearing. When ≥25% of the canola was in bloom on test fields (determined by visual estimation), colonies were moved in. The presence of 25% canola bloom ensured that bees would not forage off site, as would occur if colonies were moved to fields before bloom. Colonies were moved by pick-up truck into the canola fields during the nights of 25–26 (16 colonies), 26–27 (16 colonies), and 27 (8 colonies) June, 2012. Colonies were randomly assigned to fields. The first full day colonies were in canola fields was designated Day 0. Four colonies were positioned in the central clearing of each field so that the entrances of the colonies faced NW, NE, SE and SW.

Colonies were removed from study fields after 14 days and transported during darkness on the nights of 10–11, 11–12, and 12–13 July, 2012 to an isolated apiary. It was intended that at least 25% of canola blooms would be remaining in the fields at the time colonies were moved out of the canola fields, in order to minimize foraging of bees off site. However, due to unusually high daily temperatures and drought conditions, some canola fields were below 25% bloom at the time of colony removal. The isolated apiary was located at the Land Forces Central Area Training Facility (LFCATF) (Meaford, Ontario; 44°39′13.6″N, 80°40′52.9″W), a Canadian Forces military base approximately 165 km northeast of Guelph. So far as we are aware, this site was isolated from any crops grown from seeds treated with neonicotinoids by approximately 10 km. At the LFCATF apiary, colonies from control fields were separated from colonies from clothianidin-treated fields by approximately 40 m, and intra-treatment colonies were approximately 2 m from each other. No other colonies were present at or near to the LFCATF apiary. At this site bees foraged on a variety of wildflowers.

In September 2012, after bloom of agricultural crops in southern Ontario was finished, colonies were again prepared and moved at night from the LFCATF apiary to a winter

apiary located at the University of Guelph - Arkell Agricultural Research Station. Once at the fall/winter apiary, all colonies received medication (e.g., antibiotics), and treatments for Varroa mite and tracheal mites, based on results of the fall pest and disease assessments. Beginning 2 October 2012, colonies were fed via hive-top feeders. Each feeder was initially filled with approximately 2 L of sugar solution (2:1 sugar/water). The feeders were checked at intervals of approximately 3–4 days and refilled as needed. Colony bottom entrances were reduced and an upper entrance was provided, in mid-October. Feeders were all removed 26 October and on 15 November colonies were wrapped with insulation for protection against subfreezing temperatures.

## Colony endpoint measures

### Weight gain

Using a tripod-mounted, certified scale, colonies were weighed after dark on the night they were transported to canola fields and again after dark on the night of transport to the LFCATF apiary.

### Honey yield

Honey yield per colony was determined by weighing empty honey supers containing empty frames with foundation before placement on colonies and weighing them again after removal from colonies. Honey supers were labeled to allow cross-referencing to the colony from which the super was removed. Supers were removed from colonies when full of honey and replaced with empty supers as needed throughout the study. The sum of all honey super yields for a given colony equaled the total honey yield for that colony over approximately a 3.5-month period.

### Adult mortality

Colony adult mortality was measured in each hive using drop zone dead bee (DZDB) traps (*Rogers, Williams & Bins, 2009*), consisting of a 50 × 100 cm wood frame with fine mesh wire screening on the bottom and coarse mesh on the top, positioned at the hive entrance. The DZDB trap was a modification of a trap originally described by *Porrini et al. (2003)*. Dead worker and drone bees were removed from the traps and counted twice weekly during the period colonies were in the study fields. Collections were made early in the week and again late in the week so that duration between collections was 3 or 4 days. If available after counting, for one dead bee assessment per week, approximately 10 g of collected dead bees from each colony were pooled by field to produce a 40 g sample, and then placed in a brown glass jar, labeled, and stored frozen at $\leq -10\,°C$. Bee samples were later shipped to the USDA-APHIS National Research Center, Gastonia, NC for clothianidin residue analysis.

### Brood assessments

Brood assessments were conducted on Day-4, prior to movement of colonies to the canola fields, and at least twice while colonies were in canola (Days 7 and 14). In addition, assessments were conducted approximately every 21 days at the fall and winter apiaries until mid-October, and again in the spring of 2013.

During each colony assessment, presence or absence of eggs and unsealed larvae was determined visually and noted. The number of sealed brood cells on all frames was determined for all colonies. After doing adult strength assessments (see below), adult bees were shaken and brushed off frames into the brood box. The number of sealed brood cells on 9 frames per colony (i.e., two sides per frame) was captured as digital images using Canon EOS 5D Mark II digital cameras with 100 mm Macro lenses and portable fabric light-boxes, which facilitated consistent lighting during the image collections. A camera-mounting device that also contained a pivoting frame rack was placed into the light box. Colony frames were set on the rack while images were recorded. This mounting device facilitated an exact focal length for every digital image collected. The digital cameras were computer-controlled (laptop PC) and collected images were automatically stored on the camera memory card and simultaneously downloaded to the laptop PC hard drive in the field as a back-up precaution. Collected digital images were downloaded to at least one secondary electronic data storage medium once the equipment was returned to the laboratory.

Digital images were analyzed using IndiCounter®, Version 2.3, digital image analysis software (WSC Scientific, Heidelberg, Germany). The analysis software counted the number of sealed brood cells per frame. High accuracy of the counting software was verified by comparing values obtained on 100 randomly selected images (frames) to manual counts of cells from those images; the difference in counts with the software and manual counts was an average of $\pm 1.00\%$ (42 cells per frame; $t$-test, $P = 0.82$). The quantified values for number of sealed brood cells per frame were used to calculate total sealed brood cells per colony.

### Adult strength assessments

Adult strength assessments were conducted and verified concurrently with the brood production assessments using the methodology and equipment described for sealed brood. Digital images were acquired with adult bees present on both sides of each brood frame in each colony. The raw images were transferred to a laptop PC in the field and copied again to a second data storage medium in the laboratory. The IndiCounter software identified and counted individual bees on each frame and these numbers were used to calculate total number of bees in the hive at the time of the assessment.

### Spring 2013 colony assessments

Between 20 and 25 May 2013, when temperatures were $\geq 15$ °C and there was no heavy rainfall, the following data were collected: determination of dead and live colonies; capped brood assessment with digital imagery; adult strength assessment with digital imagery; determination of presence of queen, eggs and larvae; beeswax for residue analysis; and bee samples for Varroa mite, tracheal mite, and *Nosema* spore counts. Methods used were as described above.

## Sample collection

Nectar, honey, pollen, and beeswax were collected from colonies at each field (samples from four colonies pooled by field) on Day-4 (except pollen), 7, and 14, seven days after

movement of colonies to the LFCATF apiary, and thereafter at approximately 21-day intervals. Nectar and honey were collected until mid-October, and the final pollen samples were taken on 18 September. The final beeswax samples in autumn 2012 were collected 21 September, and beeswax samples were again collected on 20–25 May during the spring 2013 colony assessment. Collections on Day 84 occurred over 3 days (17, 18, 21 September, 2012) due to inclement weather.

Nectar that was freshly deposited in wax cells of the brood box or honey supers, was extracted using a new disposable pipette or syringe, or by gently shaking a brood frame over a large piece of wax paper and pouring the expelled nectar off the paper into a labeled brown Nalgene or glass jar (5 g samples). Honey (5 g samples) from capped cells was collected using new disposable spatulas or syringes placed in a labeled brown Nalgene or glass jar. Areas of approximately 3 $cm^2$ of food-free beeswax were collected from honey supers and placed in labeled capped Nalgene vials or 50 ml centrifuge tubes with screw caps. Pollen was collected using ANEL STANDARD® pollen traps. On each collection day, the traps were active for 24 h prior to collection. Pollen samples from each hive on each collection day were separately placed in labeled sealable plastic bags, and subsequently transferred to labeled brown Nalgene or glass jars in the laboratory. For each date, equal portions of pollen from each hive were combined to make a pooled sample of at least 15 g, 10 g of which was used for pesticide residue analysis (including enough for back-up samples), with the remaining 5 g of pollen used for floral source analysis. When in the field, all samples were immediately placed on ice or frozen ice substitute in a cooler, and placed in a freezer at $\leq -10\,°C$ when returned to the laboratory the same day.

## Pollen source analysis

Subsamples of pollen collected from pollen traps were used to determine the percent composition, by flower type, of the pollen collected by honey bees when in canola fields and when in the LFCATF apiary site. Flower samples from flowering crops or wild flowers observed in the vicinity of the study fields were photographed, collected and dried in small, labeled, paper envelopes periodically during the study. The floral samples and photographs were used as reference checks for the pollen analysis.

## Residue analysis

Nectar, pollen, honey, beeswax, and dead bee samples, previously frozen, were packed on frozen gelpacks and delivered to the USDA-APHIS National Science Laboratory, Gastonia, NC for analysis. Residue analysis for pollen, honey, nectar and beeswax was initially performed using a broad pesticide screening method (LOD for clothianidin = 1.5 ppb). Because agricultural commodities have complex matrices that can interfere with analytical procedures for detecting pesticide residues or other analytes, an extraction procedure was used to improve the detection of pesticide residues. Samples were extracted for pesticide residue analysis using method AOAC2007.01 (*AOAC, 2007*). This method utilizes the QuEChERS (Quick, Easy, Cheap, Effective, Rugged, and Safe) approach to reduce sample suppression or enhancement effects that matrices may create during chromatographic analysis. Analytes of interest were extracted from samples by high-speed grinding in

an acidified acetonitrile and water mixture followed by a "clean-up" to remove some matrix components and filtration to remove particulates. Separate aliquots of extract were analyzed for pesticide residue by gas chromatography (GC) and liquid chromatography (LC) techniques utilizing mass selective detection systems. A total of 70 honey and nectar samples, 80 beeswax samples, 20 dead bee samples, and 60 pollen samples were analyzed using this method.

After the screening analyses were complete, personnel at the Gastonia USDA lab analyzed back-up aliquots of the same nectar and pollen samples using an analytical method specifically for detecting clothianidin residues (LOQ = 1.0 ppb, LOD = 0.6 ppb). To improve detection sensitivity for clothianidin, extraction procedures were used according to *Kamel (2009)*. Analytes of interest were extracted from samples by high-speed grinding in a mixture of high purity acetonitrile, water, and triethyl amine followed by a "clean-up" procedure. Separate aliquots of extract were analyzed for clothianidin and metabolite residues by LC techniques utilizing mass selective detection systems.

Any nectar and pollen sample materials remaining, after the two USDA analyses, were transported to Bayer CropScience (BCS) in Research Triangle Park, North Carolina, where they were again analyzed for presence and quantitation of clothianidin residues using a more sensitive analytical method (LOQ = 0.5 ppb; LOD = 0.35 ppb) (*Billian & Schoning, 2009*).

### Data analysis

*t*-tests were conducted to compare the effect of clothianidin seed-treatment on levels of certain mites and diseases, colony weight gain, honey yield, overall pollen collection, and overwinter survival. Data on the number of dead bees, adults, sealed brood cells, and residues in pollen were analyzed using repeated measures multivariate analysis of variance using the standard least squares fit model platform (Manova) in JMP (*SAS, 2012*) with treatment as the fixed effect and time as the repeated (random) effect. Assumptions of normal distribution of the error term and homogeneity of variance were met for all analyses. For these data, pseudo-replication was avoided by using a single datum (mean of the sub-samples) for each experimental unit (*Hurlbert, 1984*; *Whitlock & Schluter, 2009*). Spring 2013 *Nosema* spore count data were analyzed using a multivariate standard least squares model incorporating fixed factors of treatment and colony survivorship (i.e., colonies that were classified as dead or alive). *Nosema* data were square-root transformed before analysis to fulfill normality assumptions. Unless stated otherwise, values are presented as means ± standard deviation. All data analyses were done using JMP software (*SAS, 2012*).

## RESULTS

### Pests and diseases 2012

Counts of Varroa mites were low in our colonies. There was no difference in Varroa mite levels of control and treatment colonies before exposure to clothianidin, and although the number of mites per 100 bees increased while in canola fields, there was no effect

**Table 1 Effects of exposure to clothianidin seed-treated canola fields ($n = 5$) on various honey bee colony endpoints.** Statistically significant effects ($\alpha = 0.05$) are highlighted in bold.

| Endpoint | Effect measure | Statistics |
|---|---|---|
| **SUMMER 2012** | | |
| Initial colony weight (kg) | Treatment | $t_8 = 1.05, P = 0.32$ |
| Weight gain in canola (kg) | Treatment | $t_8 = -0.18, P = 0.87$ |
| Honey yield (kg) | Treatment | $t_8 = 0.21, P = 0.84$ |
| Total pollen collected (g) | Treatment | $t_8 = -1.63, P = 0.17$ |
| Varroa mites per 100 bees | Treatment | $F_{1,8} = 0.088, P = 0.77$ |
| | Time | $F_{1,8} = 15.54, \boldsymbol{P = 0.0043}$ |
| | Treatment * Time | $F_{1,8} = 0.60, P = 0.46$ |
| No. dead bees (per 4 days) | Treatment | $F_{1,8} = 0.062, P = 0.80$ |
| | Time | $F_{3,6} = 11.29, \boldsymbol{P = 0.007}$ |
| | Treatment * Time | $F_{3,6} = 2.94, P = 0.12$ |
| No. adults | Treatment | $F_{1,8} = 0.24, P = 0.20$ |
| | Time | $F_{6,3} = 2.30, P = 0.26$ |
| | Treatment * Time | $F_{6,3} = 3.12, P = 0.19$ |
| No. sealed brood cells | Treatment | $F_{1,8} = 0.001, P = 0.92$ |
| | Time | $F_{6,3} = 9.35, \boldsymbol{P = 0.047}$ |
| | Treatment * Time | $F_{6,3} = 0.73, P = 0.66$ |
| % canola pollen collected by bees | Treatment | $F_{1,8} = 0.55, P = 0.47$ |
| | Time | $F_{1,8} = 9.89, \boldsymbol{P = 0.014}$ |
| | Treatment * Time | $F_{1,8} = 0.18, P = 0.69$ |
| Amount of pollen collected daily (g) | Treatment | $F_{1,8} = 2.64, P = 0.14$ |
| | Time | $F_{5,4} = 6.80, \boldsymbol{P = 0.044}$ |
| | Treatment * Time | $F_{5,4} = 0.93, P = 0.54$ |
| Pollen clothianidin residues | Treatment | $F_{1,8} = 7.62, \boldsymbol{P = 0.025}$ |
| | Time | $F_{1,8} = 0.60, P = 0.46$ |
| | Treatment * Time | $F_{1,8} = 2.81, P = 0.13$ |
| **SPRING 2013** | | |
| Overwinter mortality | Treatment | $t_8 = -0.69, P = 0.51$ |
| No. adults | Treatment | $t_8 = -0.41, P = 0.69$ |
| No. sealed brood cells | Treatment | $t_8 = -0.49, P = 0.64$ |
| *Nosema* counts | Treatment | $F_{1,1} = 1.18, P = 0.29$ |
| | Dead/Alive | $F_{1,1} = 10.36, \boldsymbol{P = 0.003}$ |
| | Treatment * Dead/Alive | $F_{1,1} = 0.02, P = 0.89$ |

of treatment (Table 1). The number of mites per 100 bees was at or below threshold levels of two and three mites per 100 bees for early and late summer, respectively, as recommended by the Ontario Beekeepers' Association (*OBA, 2012*) for both control (June: $0.74 \pm 0.58$ mites/100 bees; Aug: $2.40 \pm 0.77$ mites/100 bees) and treatment (June: $0.49 \pm 0.41$ mites/100 bees; Aug: $2.97 \pm 2.15$ mites/100 bees) colonies.

*Nosema* counts were also low in summer 2012. Samples from most control (12/20) and treatment (14/20) colonies had no *Nosema* spores detected and the mean number of spores per bee from control ($195,000 \pm 432,450$) and treatment ($122,500 \pm 269,002$) colonies

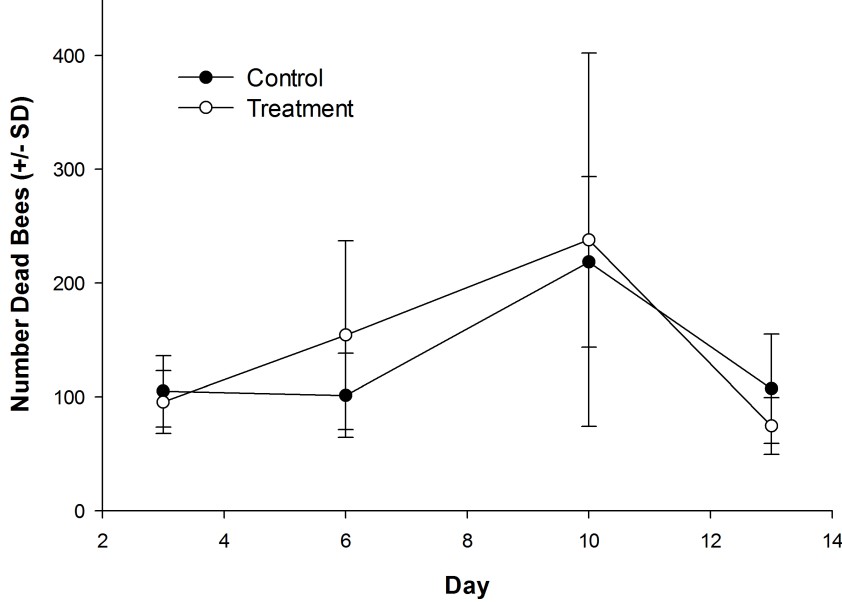

**Figure 1 Dead honey bees in front of colonies when in canola fields grown from control or clothianidin-treated seed.** Mean number of dead honey bees collected in front of colonies over 3–4 day intervals while in canola fields grown from seeds treated with or without (control) clothianidin ($n = 5$ fields per treatment).

was not significantly different ($t_8 = 0.63, P = 0.53$). None of the treated or control hives showed any presence of AFB or EFB, and incidence of chalkbrood was generally very low. In June 2012, the incidence of tracheal mite exceeded the recommended threshold of 10% infestation (*OMAFRA, 2013a*) in six control colonies and five treatment colonies. The tracheal mite threshold was not exceeded for any control or treatment colony samples taken in late July, 2012.

## Colony weight gain and honey yield 2012

There was no difference in mean weight between control ($27.9 \pm 1.7$ kg) and treatment ($28.9 \pm 1.5$ kg) colonies when initially placed in canola fields, or in weight gain when removed from fields for transport to the LFCATF apiary (control: $14.7 \pm 5.5$ kg; treatment: $14.2 \pm 4.0$ kg). There was also no difference in honey yield from colonies in control ($51.0 \pm 14.7$ kg) or treatment ($52.9 \pm 12.5$ kg) fields (Table 1).

## Number of dead bees, adults and sealed brood 2012

The number of dead bees collected in front of hives did vary over time, but was not influenced by treatment (Table 1; Fig. 1). Exposure to clothianidin seed-treated canola had no effect on the number of adults per colony, which did not change over time. The effect of time on adults was the same for both control and treatment colonies (Table 1; Fig. 2A). Similarly, the number of sealed brood cells per colony was not affected by treatment, although the number was reduced in the fall as queens ceased egg laying in preparation for overwintering (Table 1; Fig. 2B).

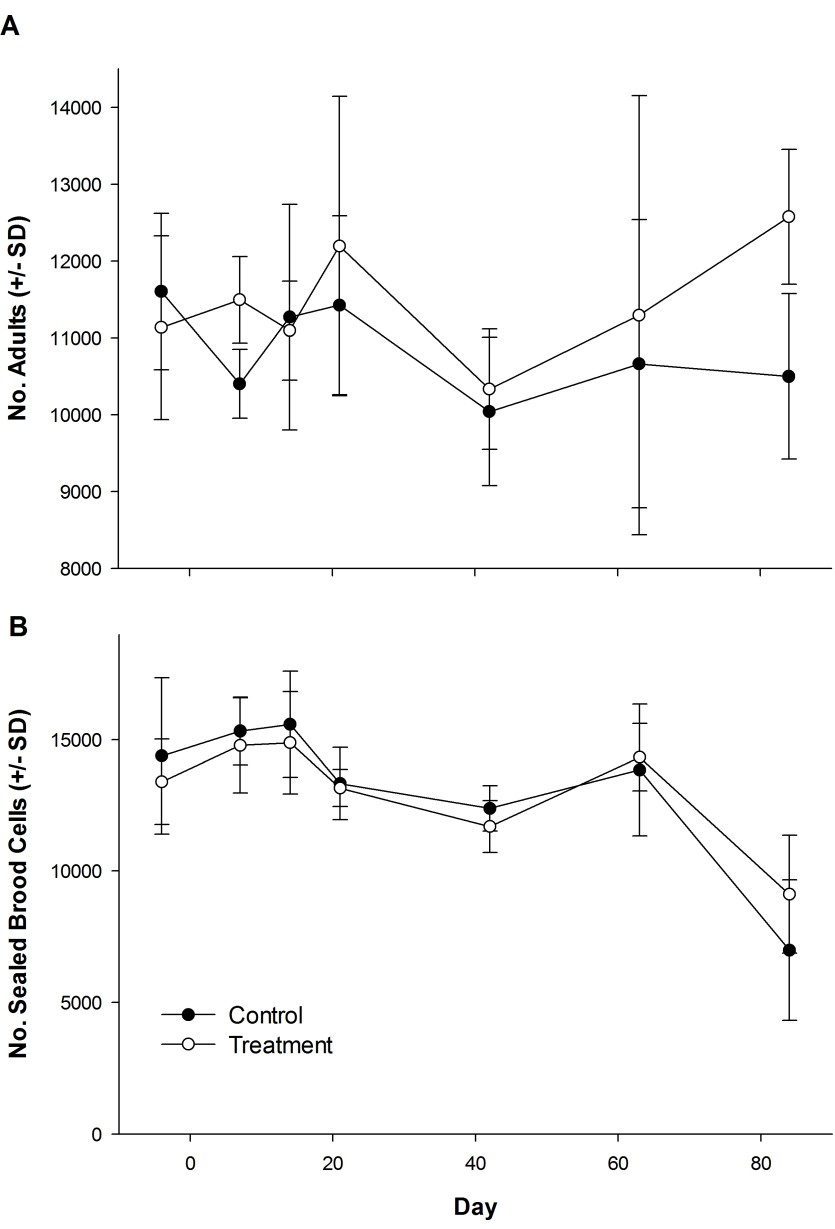

**Figure 2 Number of honey bee adults and brood during and after exposure to canola grown from control or clothianidin-treated seed.** Mean number of (A) adult honey bees and (B) sealed brood cells in colonies during and after placement in canola fields grown from seeds treated with or without (control) clothianidin ($n = 5$ fields per treatment). Colonies were in canola for 14 days and thereafter moved to an isolated apiary away from agricultural crops.

## Pollen collection 2012

Honey bees foraged heavily on canola the first week of their introduction to canola fields (Table S1). Canola pollen accounted for 88% of total pollen recovered from pollen traps on Day 7 (control: 84.9 ± 15.2%; treatment: 91.0 ± 6.2%). The amount of canola pollen collected did not differ among treatment and control fields (Table 1), but foraging on canola dropped sharply toward the end of week two (Table 1), with only 46% of the total

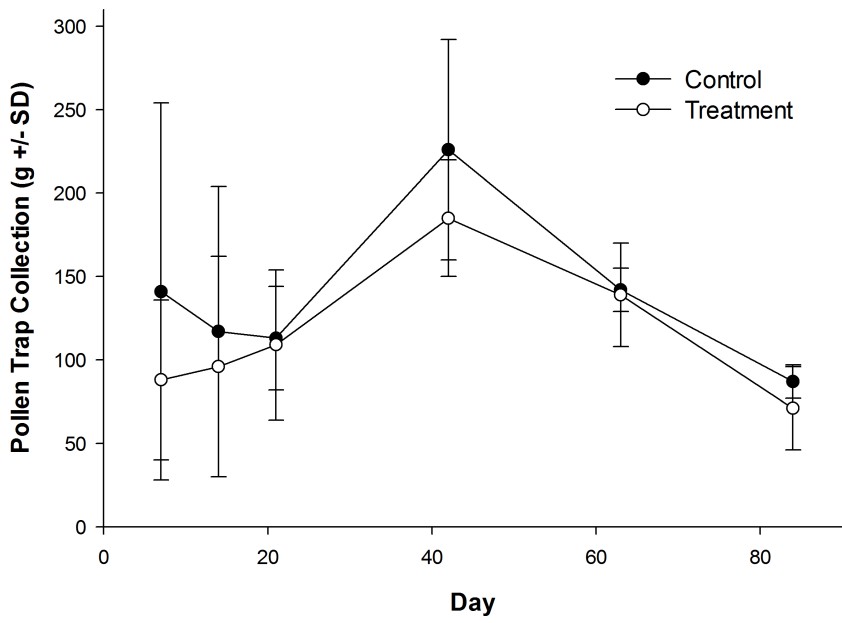

**Figure 3 Pollen recovered from honey bee colonies during and after exposure to canola grown from control or clothianidin-treated seed.** Mean amount of pollen recovered from pollen traps on honey bee colonies during and after placement in canola fields grown from seeds treated with or without (control) clothianidin ($n = 5$ fields per treatment). Colonies were in canola for 14 days and thereafter moved to an isolated apiary away from agricultural crops.

being canola pollen on Day 14 (Table S1; control: $37.5 \pm 43.3\%$; treatment: $54.8 \pm 46.0\%$). All other pollen recovered in pollen traps was from wild flowers or ornamentals (Table S2) with the exception of corn (*Zea mays* L.), which was recovered from some colonies in control and treatment fields in small amounts in week two (Day 14; range $= 0$–$7\%$ of total pollen content; mean $= 1.3\%$ of total pollen content). Field treatment also had no effect on daily pollen collection per colony during or after their placement in canola fields (Fig. 3), or on the total pollen collected from hives during the experiment (Table 1; control: $827 \pm 187$ g; treatment: $688 \pm 41$ g). There was a significant effect of time on the amount of pollen collected (Table 1), with an increase in pollen recovered from pollen traps on day 42 when colonies were in the LFCATF apiary (Fig. 3). After honey bee colonies were removed from canola fields and placed in the LFCATF apiary, no canola, corn, or soybean pollen, nor pollen from any other crop, was recovered from pollen traps.

## Spring assessment 2013

One colony from a control field and one from a treatment field died prior to overwintering. In both cases the queen disappeared from the colonies. After several weeks of monitoring, the colonies remained queenless, with no supercedure cells, no eggs and no larvae. These conditions defined a "dead" colony. Of the 19 control colonies that were alive going into winter, 7 were classified as "dead" the following April (37% overwinter colony loss). Fewer colonies from treatment fields died over the winter (5 of $19 = 26\%$ overwinter colony loss), but clothianidin seed-treatment had no statistically significant effect on percent colony

mortality (Table 1). Two additional control colonies, and one additional treatment colony, which were noted to be weak in April, died between the April assessments and the final colony assessments made during 20–25 May.

Among live colonies, there was no difference in the number of adults (control: 8,069 ± 5,317 individuals; treatment: 6,834 ± 4,005 individuals) or capped brood cells (control: 6,438 ± 5,657 cells; treatment: 4,968 ± 3,617 cells) in spring 2013 (Table 1). Varroa mite counts were very low, with less than one mite per 100 bees detected in all colonies. Incidence of tracheal mite was low. There was a single control colony with a 2% infestation level, and a single treatment colony with a 2% infestation; all other colonies had no tracheal mite detected. AFB and EFB were not detected in any colony, and chalkbrood was only detected on two frames of a single hive. Low incidence (3–4 larvae) of wax moth (*Galleria mellonella* L.) larvae was detected in two colonies.

Because no live bees could be sampled from dead colonies, *Nosema* analysis was conducted on dead bees from dead colonies, whereas *Nosema* analyses for living hives were conducted on live bees. Whether colonies were from treated or control fields had no effect on the number of spores per bee, but the number of *Nosema* spores recovered from bees from dead colonies ($2.2 \times 10^7 \pm 2.5 \times 10^7$ spores/bee) was almost a full order of magnitude higher than *Nosema* levels in live colonies ($4.9 \times 10^6 \pm 7.0 \times 10^6$ spores/bee) (Table 1). Spore counts in live control colonies ($6.5 \times 10^6 \pm 8.3 \times 10^6$ spores/bee) were similar to that in live treatment colonies ($3.5 \times 10^6 \pm 5.6 \times 10^6$ spores/bee) ($t_{24} = -1.20, P = 0.24$).

## Residue analysis

Analysis of nectar, honey, and beeswax samples by the USDA-APHIS National Science Laboratory resulted in no detection of clothianidin in these matrices (LOQ = 1.0 ppb; LOD = 0.6 ppb). The USDA Laboratory analysis of pollen collected from pollen traps detected quantifiable levels of clothianidin in only one sample from control fields (1.5 ppb) and one sample from treatment fields (1.1 ppb). Trace amounts less than the LOQ (1.0 ppb) were detected in one other control pollen sample, and one other treatment pollen sample. Detections from treatment fields were from samples collected the first week colonies were in canola, whereas clothianidin detections in control fields were from samples collected from colonies during the second week.

Enough pollen sample material was available to have the BCS Residue Analysis Laboratory analyze pollen samples with a more sensitive method (LOQ = 0.5 ppb; LOD = 0.35 ppb). There were no detections of clothianidin in pollen collected from traps seven days after placement of hives in control fields (0 detections from 5 samples), but samples collected at this time from each of the five treatment fields had quantifiable clothianidin residues at levels of 0.6, 0.8, 1.0, 1.1, and 1.1 ppb. For pollen samples collected 14 days after placement in canola fields, quantifiable residues of clothianidin were found in four of five treatment samples (0.5, 0.6, 0.8, and 1.9 ppb), and two of five control samples (0.5 and 1.3 ppb). One additional Day 14 control sample had a detectable, but unquantifiable residue of clothianidin (0.38 ppb). Analyses of samples from all matrices collected after colonies were moved out of canola fields to the LFCATF had no detections of clothianidin.

Over the two weeks of exposure of colonies in canola fields, the amount of clothianidin in the pollen from control colonies was significantly lower than that from colonies in treatment fields, but there was no significant effect of time or the treatment-time interaction (Table 1). Over both sampling periods, mean clothianidin residues in colonies from control fields ($0.24 \pm 0.44$ ppb) were over 3-fold lower than residues in colonies from treatment fields ($0.84 \pm 0.49$ ppb).

Although a number of other pesticides were detected in various matrices from control and treatment hives, most of the 173 pesticides included in the broad screen conducted by the USDA-APHIS National Science Laboratory were not detected (Table 2). The acaricides coumaphos (or its oxon), fluvalinate, 2,4-dimethylphenyl formamide (the main metabolite of amitraz), and thymol, and the fungicide chlorothalonil, were detected relatively often in beeswax, and far less often in nectar, pollen, honey, and dead bees. Other pesticides were detected rarely or in trace amounts (below the LOQ) (Table 2).

## DISCUSSION

The study results suggest that exposure to canola grown from clothianidin-treated seed had no adverse effect on honey bee colonies. There were no significant differences between colonies placed at treatment sites in comparison to control sites for hive weight gain and honey production. Our average honey yields of 50+ kg/colony (produced over a 3.5 month period) were higher than the 2012 (37.2 kg) and 5-year (37.7 kg) honey yield averages (produced over a 5–6 month period) for Ontario (*OMAFRA, 2013b*). Considering the normal turn-over rate of bees in a healthy colony (*Winston, 1987*), and high recovery rate of dead bees previously recorded with DZDB traps (*Rogers, Williams & Bins, 2009*), the number of dead bees we recorded in front of hives in this study was low and normal. Likewise, adult strength (number of adult bees) and amount of sealed brood over the course of summer and autumn 2012 and spring 2013 did not differ between treatments. With the exception of one control and one treatment colony that died during the summer (likely as a result of queen loss, which is not unusual given the intense data collection and transport of the colonies), all colonies performed very well during the summer and autumn.

Overwintering success likewise did not differ significantly between treatment and control colonies. Winter colony loss rates were higher than expected, at 37% for control and 26% for treatment colonies, but overall (32%) were similar to overwintering colony loss rates reported for the winter of 2012–2013 for beekeepers in Ontario (38%) and Canada as a whole (29%) (*CAPA, 2013*). Disease incidence was low during the summer, and Varroa mite levels were low for the duration of the study. However, 10-fold more *Nosema* spores were detected in bees from dead colonies than live colonies in spring 2013. Although we did not measure *Nosema* loads in dead bees from colonies that survived overwinter, these results suggest that there may be a correlation between overwintering survival and *Nosema* infection in our experiment. Colonies with high infection of *N. apis* may not survive winter, and those that do typically have poor spring build-up (*Pernal & Clay, 2013*). We did not observe high levels of other pests, diseases, or viruses that

**Table 2 Pesticides in honey bee colony matrices during and after exposure to canola grown from control or clothianidin-treated seed.** Pesticide detections from different matrices during and after placement of honey bee colonies in canola fields grown from clothianidin-treated seed, or untreated seed. A total of 70 honey and nectar samples, 80 beeswax samples, 60 pollen samples, and 20 dead bee samples were analyzed. Detections reported at or above the limit of quantification are presented.

| Pesticide | Pollen | | | Nectar | | | Honey | | | Beeswax | | | Dead bees | | |
|---|---|---|---|---|---|---|---|---|---|---|---|---|---|---|---|
| | # positive (%) | mean | max | # positive (%) | mean | max | # positive (%) | mean | max | # positive (%) | mean | max | # positive (%) | mean | max |
| Azoxystrobin[b] | 2 (3.3) | 17.8 | 18.6 | 0 | – | – | 0 | – | – | 0 | – | – | 0 | – | – |
| Captan[c] | 0 | – | – | 0 | – | – | 0 | – | – | 0 | – | – | 0 | – | – |
| Chlorothalonil[c,d] | 0 | – | – | 0 | – | – | 0 | – | – | 27 (33.8) | 533 | 1,060 | 1 (5.0) | 179 | 179 |
| Chlorpyrifos[c] | 0 | – | – | 0 | – | – | 0 | – | – | 0 | – | – | 0 | – | – |
| Coumaphos | 6 (10.0) | 4.1 | 8.2 | 5 (7.1) | 2.4 | 3.8 | 10 | 7.2 | 54.8 | 69 (86.3) | 133 | 1,990 | 6 (30.0) | 3.3 | 5.9 |
| Coumaphos oxon | 0 | – | – | 0 | – | – | 1 (1.4) | 7.6 | 7.6 | 69 (86.3) | 30 | 123 | 0 | – | – |
| 2,4 Dimethylphenyl formamide (DMPF) | 0 | – | – | 0 | – | – | 0 | – | – | 14 (17.5) | 39.3 | 87 | 0 | – | – |
| Fluvalinate | 0 | – | – | 0 | – | – | 0 | – | – | 8 (10.0) | 15.4 | 26.9 | 0 | – | – |
| Metolachlor[c] | 0 | – | – | 0 | – | – | 0 | – | – | 0 | – | – | 0 | – | – |
| Pendimethalin[c] | 0 | – | – | 0 | – | – | 0 | – | – | 0 | – | – | 0 | – | – |
| Propanil | 0 | – | – | 0 | – | – | 0 | – | – | 1 (1.3) | 99.7 | 99.7 | 0 | – | – |
| Propetamphos | 0 | – | – | 0 | – | – | 1 (1.4) | 12.1 | 12.1 | 0 | – | – | 0 | – | – |
| Quinoxyfen | 0 | – | – | 1 (1.4) | 18.6 | 18.6 | 0 | – | – | 0 | – | – | 0 | – | – |
| Thymol[c,d] | 1 (1.7) | 153 | 153 | 0 | – | – | 0 | – | – | 8 (10.0) | 305 | 769 | 0 | – | – |
| Trifluralin[b,c] | 0 | – | – | 0 | – | – | 0 | – | – | 0 | – | – | 0 | – | – |
| Vinclozolin[d] | 0 | – | – | 0 | – | – | 0 | – | – | 3 (3.8) | 13.6 | 23.3 | 0 | – | – |

Detection and mean or max concentration (ppb)[a]

**Notes.**

[a] Mean and max values for positive detections.
[b] Unquantifiable trace amounts detected in ≥1 dead bee samples.
[c] Unquantifiable trace amounts detected in ≥1 pollen samples.
[d] Unquantifiable trace amounts detected in ≥1 beeswax samples.

cause obvious external symptomologies (e.g., deformed wing virus, sacbrood) in control or treatment colonies during our 2012 or spring 2013 assessments. *Currie, Pernal & Guzmán-Novoa (2010)* suggested that direct and indirect effects associated with failure to control Varroa mites is the main cause of increased rates of winter colony losses in Canada, but that weather, fall feeding management, presence of *Nosema* spp., viruses and other diseases, and spring build-up of colonies, also contribute to high overwinter mortality.

Pollen trapped at hive entrances revealed a high use of the canola study fields by foraging bees. The percentage of canola pollen in traps was high (88% during peak bloom) and there was no other canola available within 10 km of each study field. During week two, the use of some of the study fields by pollen foragers, as indicated by the proportion of canola pollen collected in traps, declined sharply. This is not unexpected as honey bees have complex diet requirements (*Haydak, 1970*) and as generalists are known to utilize a wide variety of pollen and nectar sources (*Winston, 1987*). Workers expand their foraging range as they become more familiar with their surroundings, and can rapidly change their foraging patterns in response to changes in colony pollen requirements, with old floral patches being abandoned for new more favored floral resources as they are discovered (*Seeley, 1985*; *Visscher & Seeley, 1982*; *Winston, 1987*).

Residue analysis indicated that honey bees were exposed to low levels (0.5–1.9 ppb) of clothianidin in pollen. These amounts are comparable to clothianidin residue levels detected in pollen from seed-treated crops in other studies (*Blacquière et al., 2012*; *EFSA, 2012a*). These levels would not be expected to cause adverse effects based on the previously confirmed No Observable Adverse Effects Concentration (NOAEC) of 20 ppb (*Schmuck & Keppler, 2003*). We did not detect clothianidin residues in nectar, honey, or beeswax. Several other studies have reported clothianidin residues in these matrices when honey bee colonies were placed in or adjacent to clothianidin seed-treated canola. However, residues levels in these matrices are generally lower than those detected in pollen, and often residues are not detected at all in these matrices (*Blacquière et al., 2012*; *Cutler & Scott-Dupree, 2007*; *EFSA, 2012a*; *EFSA, 2012b*; *Mullin et al., 2010*; *Pilling et al., 2013*; *Walters, 2013*).

Extensive efforts were made to isolate control sites from treatment sites, by locating fields at least 10 km from each other. This was done to avoid movement of foragers between treatment and control fields, which we experienced in a previous experiment (*Cutler & Scott-Dupree, 2007*). Nevertheless, low levels of clothianidin were detected in pollen samples collected toward the end of the bloom (Day 14) from control sites 2, 3, and 6. The source of clothianidin in pollen from these colonies is unclear. Given the distance between experimental fields, it is highly unlikely that bees from control fields foraged in our treated fields (*Winston, 1987*). It also seems highly unlikely that residues in pollen were the result of carry-over in soil from previous years; if this were the case, we would have expected to find clothianidin residues in week 1 control pollen as well. All control sites were planted before treatment sites, so there is no possibility of residues being picked up on or dislodged from the seeding equipment. Control and treatment seeds are also easily distinguished by color, and our records show no mix-up occurred during planting. There is also no indication in our records of contamination or mix-up during sample collection.

Clothianidin detections from control colonies may have been a result of bees foraging off-site during the end of canola bloom. This hypothesis is supported by the fact that other fungicides and insecticides not used in our experiment were detected in colony matrices (Table 2). Analysis of pollen trap contents showed that bees continued to forage at a high rate on canola in week two at control site 6 (86% canola pollen; down from 98% canola pollen in week one at this site), but control sites 2 and 3 in week two only had canola pollen percentages of 1%, and 15%, respectively. This indicates a substantial amount of off-site foraging at these sites was occurring by the end of week two. Samples from control sites 2, 3, and 6 had very low proportions of corn pollen (5.0%, 0%, and 0%, respectively), and soybean pollen was not found in any of the pollen samples. Thus, it seems unlikely that the source of clothianidin was from pollen of corn and soybean. The vast majority of pollen from sites 2 and 3 during week 2 was from wild or ornamental plants, and these pollens may have been contaminated with clothianidin via sprays of thiamethoxam. Clothianidin is the major break-down product of thiamethoxam, and soil applications (transplant-drip) or foliar sprays of thiamethoxam can result in detections of clothianidin in pollen and nectar (*Dively & Kamel, 2012*). Actara® 25WG (25% thiamethoxam) is registered in Ontario for use against insect pests on a wide range of tree fruits, berries, and vegetables. It is possible that sprays of thiamethoxam drifted on to plants, which were subsequently foraged upon by bees from our control colonies. Irrespective of the source of clothianidin in pollen from our control colonies, our results illustrate the difficulty of conducting a perfectly controlled field study with free-ranging honey bees in real-world agroecosystems. This is especially true when conducting experiments with neonicotinoids, which are now widely used on a large number of crops and commodities.

In summary, all colonies performed well during and after the exposure period, and had overwintering success similar to colonies in Ontario and Canada on the whole. Although various laboratory studies have reported sublethal effects in individual honey bees exposed to low doses of neonicotinoid insecticides, the results of the present study suggest that foraging on clothianidin seed-treated crops, under realistic conditions, poses low risk to honey bee colonies. Our results are not conclusive as low concentrations of clothianidin were detected in some control pollen samples, but the results are consistent with those of two previous honey bee field studies with clothianidin seed-treated canola (*Cutler & Scott-Dupree, 2007*; *Scott-Dupree et al., 2001*). All three studies have shown that honey bee colonies placed during bloom in or next to canola fields grown from clothianidin-treated seeds perform as well as colonies in fields not treated with clothianidin, and as well as what is typical for honey bee colonies in Ontario. The results are also in agreement with semi-field (field cage, Tier 2) and field studies that have found that individual bees and colonies are not adversely impacted when foraging on neonicotinoid seed-treated crops (*Nguyen et al., 2009*; *Pilling et al., 2013*; *Pohorecka et al., 2012*; *Schmuck & Keppler, 2003*; *Schneider et al., 2012*; *Tasei, Ripault & Rivault, 2001*; *Thompson et al., 2013*), and corroborate the experiences of beekeepers in western Canada who for more than a decade have been producing honey in agroecosystems dominated by clothianidin seed-treated canola.

## ACKNOWLEDGEMENTS

The floral source analysis was conducted by Johanne Parent (Laboratoire BSL, Rimouski, QC). Tracheal mite and *Nosema* analyses were conducted by Brenda Perrin (Cameron, ON). Seed treatment was led by Benjamin Eakers (BCS). Site selection, planting, and management were overseen by Keith Ardiel (BCS), and conducted by Scott Ditschun, Robyn Walsh, and Katie Caldecott. We thank Paul Kelly, Apiary Supervisor at HBRF, and his support staff for advice and technical support throughout this study. We thank the beekeepers (from Canada and the US) and provincial and federal government personnel who provided helpful suggestions on the study design during a day-long field tour and open discussion at the University of Guelph in June 2012. Roger Simonds (USDA), and Gary Christensen and Audry Miller (BCS) led the residue analysis component of this study. Data collection and field assistance from Drew Mochrie, Daniel Thurston, Hilary Little, Cam Menzies, Devon Hardy and Elaine Kennedy is gratefully acknowledged.

### Funding

Funding of all expenses for this study was through Bayer CropScience. Bayer CropScience personnel had no role in collecting or interpreting field and honey bee colony data, or in writing the manuscript. Bayer CropScience employed MS and ADM as summer students, and LB as the GLP Study Director for this study. Bayer CropScience personnel assisted in treating seeds, establishing field sites, and conducting residue analysis of back-up pollen samples. GCC and CDS-D received no financial payment, research grants, travel grants, honoraria, or gifts of any kind in conducting this research or writing the manuscript. In addition to GCC, CDS-D, and LB, personnel from Bayer CropScience, the US Environmental Protection Agency, and the Health Canada Pest Management Regulatory Agency had input into the experimental design. During an 'open tour' of the experimental sites, members of the beekeeping community and provincial honey bee specialists also provided suggestions that were incorporated into the study design.

### Competing Interests

Bayer CropScience employed MS and ADM as summer students, and LB as the GLP Study Director for this study. LB is an employee of Smithers Viscient LLC.

### Author Contributions

- G. Christopher Cutler conceived and designed the experiments, performed the experiments, analyzed the data, wrote the paper, prepared figures and/or tables, reviewed drafts of the paper.
- Cynthia D. Scott-Dupree and Larry Brewer conceived and designed the experiments, performed the experiments, contributed reagents/materials/analysis tools, wrote the paper, reviewed drafts of the paper.
- Maryam Sultan performed the experiments, analyzed the data, wrote the paper, reviewed drafts of the paper.
- Andrew D. McFarlane performed the experiments, wrote the paper, reviewed drafts of the paper.

## Data Deposition

The following information was supplied regarding the deposition of related data:
Dryad: 10.5061/dryad.td03f

## Supplemental Information

Supplemental information for this article can be found online at http://dx.doi.org/10.7717/peerj.652#supplemental-information.

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
