# Peer review of "A large-scale field study examining effects of exposure to clothianidin seed-treated canola on honey bee colony health, development, and overwintering success"

_PeerJ, doi:10.7717/peerj.652_

## Round 0.1 · original submission · Minor Revisions

One reviewer recommends accepting the paper, the other recommends acceptance with minor revision. Having read the paper myself, I agree that this paper is almost ready for publication and I have nothing substantial to add to the reviewers' helpful comments and suggestions. The minor revisions are very minor and should be straightforward for the authors to address.

I expect this paper to be of high interest to the scientific/entomological community, policy makers, and the general public. It is obviously extremely relevant to the current discussions of pollinator decline and the role of pesticides in this phenomenon.

I recommend that the authors choose to make the peer review history of this paper public because the reviews add valuable information and context. Both reviewers also indicated that they are willing to make their reviews identifiably public.

·

Basic reporting

Neonicotinoid insecticides have been blamed for large-scale losses of honey bee colonies and declines in colony health. Many of the conclusions on the dangers of neonicotinoid use have been based on laboratory studies that attempt to extrapolate laboratory results to field situations. This manuscript by Cutler et al. details the results of a large-scale field study designed to examine the effects on honey bee colony health from bees foraging on canola grown from clothianidin treated seed. The authors provide sufficient introductory and background information with regard to the significance of their study, and the data analysis, results, and discussion provide a coherent body of work suitable for publication. The manuscript is clearly written, but there are a few points and editorial changes that need to be addressed.

Experimental design

The study is well-designed and the data provide clear evidence for a lack of significant impact on colony development, health or overwintering success from foraging on canola crops grown from treated seed. The experimental approach is straight-forward and the methods are clearly described. Analytical procedures follow standard protocols and should be reproducible.

Validity of the findings

The data are robust and well controlled for a field study of this nature. Field studies with honey bee colonies are typically subject to high degrees of variability, but the field replication and number of colonies used in the study provide for statistically sound results. The conclusions are appropriate and answer the initial questions as to whether honey bee foraging on canola grown from clothianidin treated seed negatively impacts colony health.

Additional comments

Comments and editorial suggests for the authors.

Line 140 – change …in the previous 12 months, ‘had not received’ (instead of did not receive) applications of clothianidin etc. (also suggest removal of Group 4 chemistry unless you define what Group 4 chemistries are with respect to mode of action classification)

Line 170 – ‘following’ board- more usual term is ‘follower’ board

Line 174-5 – did empty honey supers contain 9 frames of drawn comb or foundation? Clarify.

Line 221 – change’ forager’ to ‘foraging’

Line 254 - containing empty frames of foundation or comb?

Line 267 – you state dead bees counts were made from day 0 but your figure 1 only shows from day 3. Were any dead bee counts made before or immediately after colonies were moved?

Lines 304-5 – the difference between software and manual counts should be expressed as a percentage, not as an average of cells per frame. 42 cells per frame out of 200 sealed cells would be ~ 20% difference, but only a 2.5% difference out of 1700 cells.

Line 308f (Adult strength assessments) – Were any estimates made of the accuracy of the counts of the numbers of bees per frame? (such as shaking and weighing the bees from individual frames)

Line 314 – spelling of IndiCounter (no ‘e’)

Lines 325ff (Sample Collection)
Line 327- pollen samples were pooled, Line 345 – pollen samples were placed in labeled sealable plastic bags. Am I corrected in assuming that the samples collected in the field were not kept separate, but pooled at that time? Was any effort made to standardize the amount of pollen from each hive. If the majority of pollen came from one hive it could bias both the residue data and the pollen source data.

Line 396 – better clarify which data were analyzed by t-tests and which by ANOVA, and suggest grouping the analysis data in Table 1 by test (place honey yield and total pollen collected with the initial wt gain and wt gain in canola)

Line 400 – suggest “using a mixed model ANOVA with repeated measures” instead of ‘repeated measures multivariate…. Also was the least squares fit model or the REML method used in JMP?

Line 406 - spore count data ‘were’ …

Line 527 – you cannot give an unquantifiable residue level as 0.38 ppb. The level should be given as less than 0.5 and greater than 0.35 ppb (your upper and lower limit values for quantification and detection). The use of the 0.38 ppb value would also affect calculations in line 534.

Line 547 – suggest change to: …that exposure to canola grown from clothianidin treated seed (from clothianidin seed-treated canola)

Line 562 – suggested change: all colonies ‘performed’ very well…

Line 573 - suggested change: overwinter, ‘these results suggest’

Line 575 – your mention N. apis but was any effort made to determine if the coloies in your study had N. apis or N. ceranae?

·

Basic reporting

The manuscript describes a coherent, self-contained study and is written to a high standard and with comprehensive reference to the relevant literature in the field. I have no major comments to make, but the authors may like to consider the following minor suggestions:

1. As the debate over the impact of neonicotinoid insecticides on bees spans several continents an international readership might be expected for this paper, some of whom will be from areas where both spring and autumn canola/oilseed rape varieties are grown. Such varieties vary in the interval between sowing and flowering (spring-sown - a few weeks; autumn-sown – up to six months) potentially leading to higher residues in pollen and nectar in spring-sown crops. Thus for the benefit of these readers it may be useful to state the sowing dates and varieties of the crops used in this experiment.
2. Line 193 (Similarly Line 540): Amitraz is described as a miticide. Although a correct term for North American readers the authors might consider replacing or adding the term “acaricide”, again for the benefit of an international readership?
3. Line 221: “to minimize forager of bees” should read “minimize foraging of bees”.
4. Line 305-307: Considering its’ potential use by other researchers in future studies, the authors might consider reporting in more detail (including graphical/tabular representation of the data) results of the comparison of number of sealed brood cells per frame when assessed by digital imaging and manual counting.
5. Line 452 et seq. The authors might consider a graphical representation of species profile of pollen collected with time to illustrate their data.

Experimental design

This is an excellent study, carefully designed to overcome many of the pitfalls that have beset too much of the published work in this field, and the authors are to be congratulated on the quality of the dataset they have produced. Having recently reviewed the published (and some unpublished) data on the effect of neonicotinoids on bees, several key problems that make many other studies in the literature difficult to interpret in terms of their relevance to the current debate can be identified. In particular, these include unrealistically high rates of exposure to insecticides that are unlikely to be encountered in the field; most studies addressing imidacloprid with relatively few on other neonicotinoid a.i.s; and assessment of endpoints whose biological significance in terms of colony performance/fitness is unknown. This study overcomes all of these, achieving (commercially and environmentally) field-realistic exposure of bees to clothianidin, an understudied member of the neonicotinoid group. The value of this work is further enhanced by the selection of a comprehensive set of endpoints to assess that collectively enable a reliable assessment of colony performance.

Currently there are widespread calls for field-based investigations of this topic but the authors correctly acknowledge (lines 652-653) the difficulties of conducting perfectly controlled field experiments on insecticides. The design adopted has addressed these constraints as far as is possible, and provide sufficient details in their report to enable the reader to reach sound and robust conclusions based on the findings. For example, the emphasis placed on careful site selection to reduce the potential for foraging on non-experimental treated crops is of key significance and will have achieved this aim. In addition, the known rates of decline of environmental residues of neonicotinoids following seed treated wheat makes the 1.5 year period before commencement of this study sufficient to ensure that no adverse effects would occur. Similarly the details of crop husbandry provided adequately reflect normal commercial practice in Ontario, and colony handling procedures were sound, adding further confidence that the results are meaningful and accurately reflect the likely impact of clothianidin in the agricultural environment.

With regard to other aspects, the Materials and Methods section reports details of a strong experimental design that supports the development of sound conclusions and results relevant to the current debate.

Validity of the findings

The sound experimental design and manner in which the study has been conducted lead to the generation of assessments of a range of colony characteristics and associated environmental residue measurements that are mutually consistent and point to logical and robust conclusions on the findings of this study. It yields no evidence of significant adverse effects of clothianidin seed-treated canola on honey bee colony performance. This reflects the findings of the very few other field studies (conducted on other neonicotinoid a.i.s) adding confidence to the findings. Studies of this type are expensive and replication within and between years is usually limited by the available funding. Although this study was conducted in only a single year, the within-year replication (number of fields used) is higher than most equivalent investigations that I am aware of, and makes the findings yet more noteworthy. The authors should seek funding for further work of this type, but publication of the current study must not be delayed whilst the outcome of funding applications are awaited.

I believe that the study has generated important and valid findings, and sound conclusions that will be of significant value in resolving the current debate on potential impact of neonicotinoid use on Apis species. The timeliness of the publication and worldwide significance is illustrated by the recent moratorium in the EU of the use of imidacloprid, thiamethoxam and clothianidin as seed treatments on bee attractive crops (including canola/oilseed rape), which has resulted in reliance on multiple sprays of old insecticide classes (and already, complete crop losses due to outbreaks of flea beetles at establishment/young plant stages). The moratorium is for two years, during which period peer reviewed publications describing well targeted/designed work such as this study are urgently needed to inform the re-appraisal of the moratorium by EU policy makers, and the ongoing legislative consideration of the issues on other continents.

Additional comments

Against the background of the current controversy on the potential impact of neonicotinoids on pollinators this is an important, well designed study that represents a significant contribution to the scientific understanding required to resolve the polarised debate over their future use. Further work of this type will be valuable and it is to be hoped that funding can be obtained to enable continuation of this research programme. Given the worldwide legislative interest in the future of this insecticide class, this paper should not only be published as rapidly as possible, but the findings actively promulgated widely to relevant stakeholders including government, academia, growers, the agrochemical industry, environmental bodies and others.

---

## Round 0.2 · accepted · Accept

Thank you for your prompt response to the reviews. As noted earlier, the reviewer-requested changes were very straightforward, and you and your co-authors have dealt with them adequately and appropriately.

Please consider making the review history for this MS public, as that added information can be extremely valuable as readers assess your results and conclusions.

External reviews were received for this submission. These reviews were used by the Editor when they made their decision, and can be downloaded below.